# Blood Flow Analysis of the Great Saphenous Vein in the Su-Pine Position in Clinical Manifestations of Varicose Veins of Different Severities: Application of Phase-Contrast Magnetic Resonance Imaging Data

**DOI:** 10.3390/diagnostics12010118

**Published:** 2022-01-05

**Authors:** Yuan-Hsi Tseng, Chien-Wei Chen, Min-Yi Wong, Teng-Yao Yang, Yu-Hui Lin, Bor-Shyh Lin, Yao-Kuang Huang

**Affiliations:** 1Division of Thoracic and Cardiovascular Surgery, Chiayi Chang Gung Memorial Hospital, College of Medicine, Chia-Yi and Chang Gung University, Taoyuan 33302, Taiwan; 8802003@cgmh.org.tw (Y.-H.T.); mynyy001@gmail.com (M.-Y.W.); vw200162@cgmh.org.tw (Y.-H.L.); 2Department of Diagnostic Radiology, Chiayi Chang Gung Memorial Hospital, College of Medicine, Chia-Yi and Chang Gung University, Taoyuan 33302, Taiwan; chienwei33@gmail.com or; 3Institute of Medicine, Chung Shan Medical University, Taichung 408, Taiwan; 4Institute of Imaging and Biomedical Photonics, National Chiao Tung University, Tainan 300, Taiwan; borshyhlin@gmail.com; 5Department of Cardiology, Chiayi Chang Gung Memorial Hospital, College of Medicine, Chia-Yi and Chang Gung University, Taoyuan 33302, Taiwan; 2859@adm.cgmh.org.tw

**Keywords:** magnetic resonance imaging, magnetic resonance venography, non-contrast venography, phase-contrast magnetic resonance imaging, quantitative flow, varicose vein, great saphenous vein

## Abstract

The progression of clinical manifestations of lower-limb varicose veins remains unclear. This study investigated changes in lower-limb venous blood flow using phase-contrast magnetic resonance angiography. Data were collected on veins from 141 legs. We compared legs with and without varicose veins and related symptoms and examined varying levels of varicose vein symptom severity. Legs without varicose veins exhibited a lower absolute stroke volume (ASV, *p* < 0.01) and mean flux (MF, *p* = 0.03) for the great saphenous vein (GSV) compared with legs with symptomatic varicose veins. Legs with asymptomatic varicose veins exhibited lower MF for the GSV (*p* = 0.02) compared with legs with symptomatic varicose veins. Among legs with varicose veins, asymptomatic legs exhibited lower ASV (*p* = 0.03) and MF (*p* = 0.046) for the GSV compared with legs that exhibited skin changes or ulcers; however, no significant differences were observed between legs presenting with discomfort or edema and legs with skin changes or ulcers, and between legs presenting with discomfort or edema and asymptomatic legs. In conclusion, in the supine position, increased blood flow rate and blood flow volume in the GSV were associated with symptomatic varicose veins and increased symptom severity.

## 1. Introduction

Lower-limb varicose veins are typically related to chronic venous insufficiency and pathologic changes to the venous walls [1,2] and can cause various symptoms, including aesthetic problems, lower-limb discomfort (such as heaviness, aching or throbbing pain, cramping, and burning sensations), edema, skin changes, ulcers, chronic wounds, and lymphedema [3,4,5]. However, the mechanism underlying venous disease progression from mild to severe remains unclear [6]. In addition, the severity of venous reflux does not correlate entirely with the severity of clinical manifestations, which can be influenced by many other factors, including age, sex, deep vein reflux, and the involved venous segment [7,8,9].

Although the gold standard for evaluating varicose vein severity is color-flow venous duplex ultrasound, the advent of three-dimensional phase-contrast magnetic resonance angiography (PC-MRA) has increased the ability to inspect venous anatomy from the feet to the pelvis, in addition to providing an easier and more intuitive method for measuring the venous blood flow rate and volume. However, the patient is required to maintain a supine position during MRA scans, which can influence the accuracy of estimations for venous reflux by removing the ability to evaluate the influence of gravitational forces on lower-limb venous reflux. Therefore, in this study, we aimed to investigate whether any associations exist between changes in lower-limb venous blood flow and the severity of clinical manifestations of varicose veins in the supine position.

## 2. Materials and Methods

The protocol for this retrospective study was approved by the Institutional Review Board of Chang Gung Memorial Hospital (CGMF IRB No. 202100680B0). The requirement for obtaining written informed consent was waived for this study due to the retrospective and anonymous nature of the patient data used.

In this retrospective study, we collected data on 214 legs from all 107 patients who underwent PC-MRA as part of a lower-extremity vascular survey conducted between October 2018 and December 2020. Legs without complete blood flow data and legs that did not contain varicose veins but exhibited symptoms similar to venous disease were excluded. Ultimately, the data from 141 legs were used in this study.

Baseline demographics and clinical characteristics included age, sex, comorbidities, and clinical inspection. The abnormal venous anatomies found in the PC-MRA were collected, and included compression of the iliocaval territory, deep vein thrombosis, dilatation of the great saphenous vein (GSV; dilation was defined as a vein diameter > 3 mm), in addition to the GSV to popliteal vein (PV) diameter ratio. The PV diameter was used to normalize the GSV diameter to decrease inherent differences in venous size among different people because deep vein dilatation is uncommon. The GSV and PV diameters were measured at the same height using the frontal view of three-dimensional, maximum-intensity projection images. Venous blood flow was analyzed using the quantitative flow (QFlow) technique, which collected the blood flow parameters, including the regurgitation fraction (RF), absolute stroke volume (ASV, as the forward flow volume + backward flow volume), mean flux (MF), and mean velocity (MV) of the external iliac veins (EIVs), femoral veins (FVs), PVs, and GSVs.

In addition to comparing the baseline demographic and clinical characteristics between patients with and without varicose veins, this study used two steps to analyze the lower-limb venous anatomy and blood flow data. The first step was to classify the legs into the following groups: (1) legs without varicose veins; (2) legs with asymptomatic varicose veins; and (3) legs with symptomatic varicose veins. The second step was to classify the legs with varicose vein by their clinical presentations into three groups: (1) no symptoms; (2) discomfort (such as pain, soreness, heaviness, cramping, or itchiness) or edema (C3 using the CEAP classification system); and (3) skin changes (C4 using the CEAP classification system) or ulcers (C5 and C6 using the CEAP classification system).

Among currently available non-contrast magnetic resonance imaging techniques, our hospital adopts triggered angiography non–contrast-enhanced magnetic resonance imaging (Philips Ingenia, Philips Healthcare, Best, The Netherlands). A sequence of three-dimensional turbo spin-echo techniques and a short tau inversion recovery protocol were used to separate venous structures from background tissues and arterial signals. Blood flow measurements were performed with the QFlow application (Figure 1). Details regarding the parameter settings used to image the lower extremities are described in our previous study [10]. Sequence parameters are presented in Table 1.

### Statistical Methods

Continuous variables are presented as the median and interquartile range, whereas categorical variables are presented as the number and percentage. To compare two independent groups, the Mann–Whitney U test was used to analyze continuous variables, and the Chi-square test or Fisher’s exact test was used to analyze categorical variables. To compare three or more independent groups, the Kruskal–Wallis test, followed by Dunn’s test for post hoc analyses, was used to analyze continuous variables, and the Chi-square test and pairwise Z test (for post hoc analysis) were used to analyze categorical variables.

All statistical analyses were conducted using SPSS for Windows (Version 20, IBM, Armonk, NY, USA).

## 3. Results

The median age of enrolled patients was 62 years (interquartile range: 53–74 years), with women forming the majority (66.4%). Patients with and without varicose veins are compared in Table 2, which indicated no significant differences in age, sex, body mass index, or comorbidities. Engorged superficial veins occurred most frequently in both legs (43.5%), followed by the right leg only (39.1%).

The first step analysis results are presented in Table 3, which shows that legs with symptomatic varicose veins had a higher rate of GSV dilatation than legs with asymptomatic varicose veins (*p* < 0.001). In addition, the post hoc analyses showed that legs with symptomatic varicose veins had higher values for the GSV to PV diameter ratio (*p* = 0.001), the GSV ASV (*p* = 0.004), and the GSV MF (*p* = 0.026) than legs without varicose veins. Legs with symptomatic varicose veins had a higher GSV MF value than legs with asymptomatic varicose veins (*p* = 0.02). However, legs without varicose veins and legs with asymptomatic varicose veins had similar GSV to PV diameter ratios and similar GSV blood flow parameters. The line plot also displays a substantial increase in the median GSV ASV and GSV MF values in legs with symptomatic varicose veins relative to both legs without varicose veins and legs with asymptomatic varicose veins (Figure 2).

Table 4 presents the analysis of legs with varying levels of varicose vein symptom severity. Although legs with no symptoms had the lowest rate of GSV dilatation (*p* < 0.001), no significant differences were observed between legs with discomfort or edema and legs with skin changes or ulcers (Fisher’s exact test, *p* = 0.73). Furthermore, post hoc analyses indicated that legs with skin changes or ulcers had higher values for the GSV to PV diameter ratio, GSV ASV, and GSV MF than legs with asymptomatic varicose veins; however, no significant differences were observed between legs with discomfort or edema and legs with no symptoms, and between legs with discomfort or edema and legs with skin changes or ulcers. Therefore, the values of the GSV to PV diameter ratio, GSV ASV, and GSV MF may increase with increased varicose vein progression (from legs with no symptoms to legs with discomfort or edema, and, finally, to legs with skin changes or ulcers). The line plot also showed that the GSV ASV and GSV MF of legs with discomfort or edema appear in a transition state between legs with no symptoms and legs with skin changes or ulcers (Figure 3).

## 4. Discussion

This study demonstrated that in legs with varicose veins, an increase in the GSV to PV diameter ratio and an increase in the GSV blood flow volume or rate was associated with an increased occurrence of related lower-limb symptoms and, potentially, more severe symptoms. Previous studies have reported that severe venous insufficiency is associated with the clinical symptoms of precapillary arteriole dilatation and abnormally elevated lower-limb arterial blood flow, which lead to high pressure in the capillary system of a standing individual [11,12]. Therefore, the degree of increase in GSV blood stroke volume and rate may be determined by GSV dilatation and the increase in precapillary arteriole blood flow, and the degree of increase may, in turn, determine the severity of related symptoms. In addition, this pathological change may be more obvious in the supine position during magnetic resonance imaging examination due to the lack of influence from gravitational force and muscle contractions on the forward blood flow. In addition, differences in the blood flow volume or rate may correspond to different degrees of pathological change.

The results of this study suggest that the extent of dilatation of the GSV may be an unreliable indicator of symptom severity. When analyzing varying levels of symptom severity in legs with varicose veins, legs presenting with discomfort or edema and legs presenting with skin changes or ulcers exhibited similar levels of GSV dilatation. We also observed that legs without varicose veins and legs with asymptomatic varicose veins had similar GSV blood flow parameters. These results could partially explain why some patients who present with horrible superficial vein engorgement and deformities in the lower limbs do not exhibit symptoms of corresponding severity. However, these results are not totally unexpected given that skin changes and ulcerations are commonly influenced by multiple complex biological mechanisms, such as the accumulation of inflammatory cells, the overexpression of growth factors, elevated matrix metalloproteinase expression, and increased cellular senescence [13,14,15,16,17]. These mechanisms are, in turn, affected by changes in venous pressure and shear stress [18]. By contrast, varicose changes in the superficial veins are typically determined by examining their dilatation (usually defined as vein diameter > 3 mm) and tortuosity but not their reflux or perforator incompetence; therefore, the extent of dilatation of the GSV may not be a central contributor to symptom severity.

Regardless of the grouping approach applied in this study, no between-group significant differences were identified for any blood flow parameters of the examined deep veins (PVs, FVs, and EIVs). Changes in varicose vein blood flow may have little to no influence on the blood flow in deep veins, suggesting that the loss of ability to modulate the balance of blood flow between the superficial and deep veins may represent another important factor. The connection between the superficial and deep veins is known to involve the perforator veins. Past studies have demonstrated that incompetent perforator veins are associated with the severity of chronic venous disease, and superficial venous reflux can deteriorate perforator vein competence [19,20]. However, superficial and deep veins may need to be viewed as two independent venous systems, despite their connection by perforator veins. Previous studies have indicated that deep venous reflux can cause skin changes or ulcers while influencing the surgical outcomes of GSV ablation independently [21,22]. In addition, the beneficial outcomes of treating incompetent perforator veins to decrease the recurrence rate of varicose veins remain uncertain [23]. Therefore, patients with chronic venous disease should be managed using a comprehensive method in which the functions of both the superficial and deep veins are evaluated simultaneously to determine possible treatments.

In the QFlow data analysis, the GSV RF did not increase with the increasingly severe clinical manifestations, which was unsurprising because magnetic resonance imaging must be performed while the patient is in a supine position, removing the influence of gravitational force and muscle contractions on lower-limb venous blood flow [24,25]. However, the findings of changes in the GSV blood flow rate or volume may compensate for this shortcoming. Defining meaningful cutoff values for assessing GSV blood flow parameters requires further study. In addition, although it is impossible to use PC-MRA as a substitute for color-flow duplex venous ultrasound, among patients who require PC-MRA to evaluate possible intra-abdominal or pelvic venous disease or compression, determining how best to interpret venous blood flow parameters measured by the postprocessing software (such as MR QFlow in our hospital) remains worthy of future exploration.

Many other factors are likely to influence symptom severity, such as the baseline characteristics of sex, age, and body mass index [26]. Patients with a history of leg injury or inflammation, deep vein thrombosis, and genetic abnormalities are also at higher risks of experiencing venous ulceration [27,28]. At the anatomical level, perforator vein incompetency and deep vein reflux can promote skin changes and ulceration [29,30]. Therefore, our findings can only explain the causes of symptom deterioration to a limited degree.

### Limitations

This study primarily used PC-MRA data collected from patients in the supine position. Generally, the severity of venous reflux is determined by the reflux duration observed in the standing position using venous duplex ultrasound. Therefore, PC-MRA results are likely to underestimate RF values. In addition, the lack of consideration of behavioral factors, family history, and doppler ultrasound evaluation in the assessment of venous hemodynamic changes is one of the limitations of this study.

## 5. Conclusions

In the supine position, increases in GSV blood flow rate and volume are associated with the development of symptomatic varicose veins and, potentially, the progression of symptom severity. Our findings may help physicians prevent and predict the progression of varicosity-related symptoms.

## 6. Patents

This project is under the reviewing process in the Taiwan Intellectual property Office. (No 109126307).

## Figures and Tables

**Figure 1 diagnostics-12-00118-f001:**
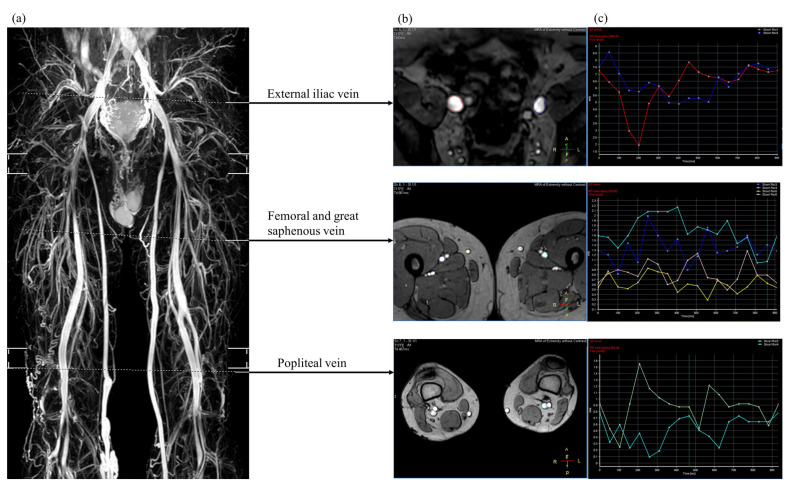
Three-dimensional, non-contrast magnetic resonance venography (**a**) showing the selected region of interest for venous blood flow analysis (**b**) and the mean flux calculated from the quantitative flow analysis within one R-R interval (**c**).

**Figure 2 diagnostics-12-00118-f002:**
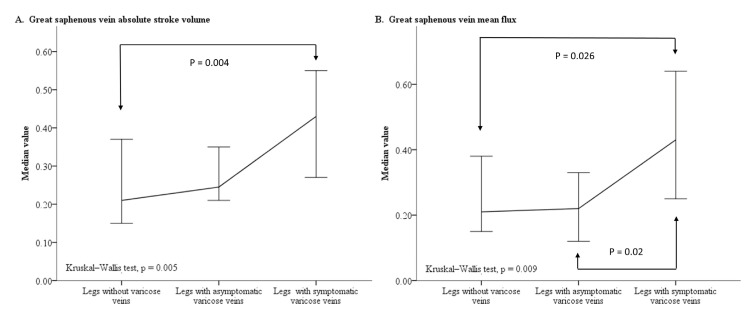
Line plot charting the median value and 95% confidence interval of the GSV ASV (**A**) and GSV MF (**B**) for legs without varicose veins, legs with asymptomatic varicose veins, and legs with symptomatic varicose veins. Abbreviations: ASV, absolute stroke volume; GSV, great saphenous vein; MF, mean flux.

**Figure 3 diagnostics-12-00118-f003:**
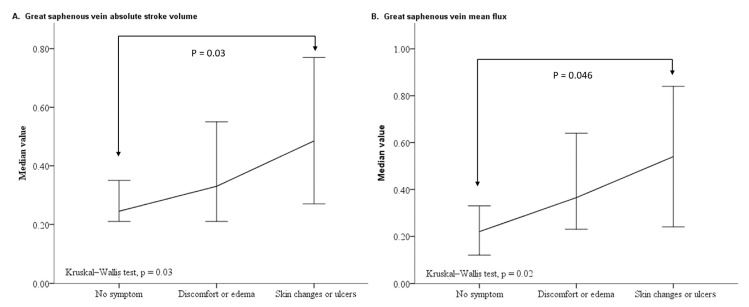
Line plot of the median value and 95% confidence interval for GSV ASV (**A**) and GSV MF (**B**) in legs with varying levels of varicose vein symptom severity. Abbreviations: ASV, absolute stroke volume; GSV, great saphenous vein; MF, mean flux.

**Table 1 diagnostics-12-00118-t001:** Parameters of triggered angiography non-contrast enhanced magnetic resonance venography.

Sequence	Parameters
Magnetic strength, T	1.5
Manufacturer	Philips
Model	Ingenia
Sequence type	Three-dimensional turbo spin echo skill with short-tau inversion recovery protocol
Phase of cardiac cycle	Systolic
Repetition time	one beat
Echo time (msec)	85
Flip angle	90°
Inversion recovery delay time (msec)	160
Voxel size (mm^3^)	1.7 × 1.7 × 4
Field of view (mm)	360 × 320

**Table 2 diagnostics-12-00118-t002:** Baseline demographics and clinical characteristics of enrolled patients.

Variables	Patients Diagnosed with Varicose Veins N = 69	Patients Diagnosed without Varicose Veins N = 38	*p*
Age (years)	61 (52.5, 72.5)	64 (54.5, 79.3)	0.22
Sex, *n* (%)			0.93
Male	23 (33.3)	13 (34.2)	
Female	46 (66.7)	25 (65.8)	
Body mass index	26.83 (24.61, 30.22)	26.80 (22.77, 30.35)	0.62
Comorbidities			
Hypertension, *n* (%)	27 (39.1)	18 (47.4)	0.41
Diabetes mellitus, *n* (%)	11 (15.9)	9 (23.7)	0.33
Hyperlipidemia, *n* (%)	13 (18.8)	7 (18.4)	0.96
History of cerebrovascular accident, *n* (%)	2 (2.9)	3 (7.9)	0.35
Chronic kidney disease, *n* (%)	3 (4.3)	2 (5.3)	1.00
Coronary artery disease, *n* (%)	3 (4.3)	3 (7.9)	0.66
Atrial fibrillation, *n* (%)	3 (4.3)	2 (5.3)	1.00
Smoking, *n* (%)			
Yes	7 (10.1)	3 (7.9)	1.00
Cessation	3 (4.3)	4 (10.5)	0.25
Chronic lung disease	4 (5.8)	0	0.29
Leg with engorged superficial vein			
Right	27 (39.1)	-	-
Left	12 (17.4)	-	-
Bilateral	30 (43.5)	-	-

**Table 3 diagnostics-12-00118-t003:** Comparison of imaging findings and venous blood flow parameters of legs without varicose vein, legs with asymptomatic varicose vein, and legs with symptomatic varicose vein.

	Legs without Varicose Veins *n* = 53	Legs with Asymptomatic Varicose Veins *n* = 42	Legs with Symptomatic Varicose Veins *n* = 46	*p*
Imaging findings				
Compression of iliocaval territory	18 (34.0)	11 (26.2)	14 (30.4)	0.72
Deep vein thrombosis	7 (13.2)	2 (4.8)	6 (13.0)	0.34
GSV dilatation	-	16 (38.1)	35 (76.1)	<0.001
GSV to PV diameter ratio *	0.47 (0.37, 0.63)	0.52 (0.43, 0.75)	0.67 (0.50, 0.79)	0.001
Blood flow analysis				
External iliac vein				
RF (%)	0.00 (0.00, 0.00)	0.00 (0.00, 0.00)	0.00 (0.00, 0.08)	0.39
ASV (mL)	2.93 (2.02, 5.10)	2.34 (1.85, 5.32)	3.50 (2.53, 4.82)	0.67
MF (mL/s)	3.29 (2.22, 5.60)	2.82 (2.01, 5.95)	3.41 (2.51, 5.35)	0.87
MV (cm/s)	5.32 (3.95, 7.26)	6.61 (4.56, 8.48)	7.51 (4.79, 9.79)	0.09
Femoral vein				
RF (%)	0.00 (0.00, 0.00)	0.00 (0.00, 0.00)	0.00 (0.00, 0.00)	0.56
ASV (mL)	1.06 (0.48, 1.55)	1.13 (0.82, 1.86)	1.11 (0.78, 1.78)	0.12
MF (mL/s)	1.10 (0.65, 1.79)	1.32 (0.94, 1.98)	1.37 (0.85, 2.08)	0.14
MV (cm/s)	3.26 (2.21, 4.75)	3.60 (2.61, 5.18)	4.60 (2.88, 6.32)	0.08
Popliteal vein				
RF (%)	0.00 (0.00, 1.54)	0.00 (0.00, 0.00)	0.00 (0.00, 0.00)	0.15
ASV (mL)	0.63 (0.35, 1.09)	0.80 (0.46, 1.10)	0.77 (0.46, 1.38)	0.26
MF (mL/s)	0.71 (0.41, 1.23)	0.85 (0.51, 1.14)	0.89 (0.44, 1.50)	0.27
MV (cm/s)	1.54 (0.97, 2.44)	1.70 (1.26, 2.31)	1.95 (1.30, 3.53)	0.13
Great saphenous vein				
RF (%) ^#^	0.00 (0.00, 10.05)	2.08 (0.00, 33.77)	0.00 (0.00, 2.15)	0.03
ASV (mL) ^†^	0.21 (0.12, 0.43)	0.25 (0.16, 0.47)	0.43 (0.22, 0.67)	0.005
MF (mL/s) ^‡^	0.21 (0.11, 0.54)	0.22 (0.08, 0.43)	0.43 (0.21, 0.84)	0.009
MV (cm/s)	1.80 (0.76, 3.03)	1.46 (0.55, 2.80)	2.43 (1.10, 4.39)	0.12

* Pairwise comparison revealed a significant difference between legs without varicose veins and legs with symptomatic varicose veins (*p* = 0.001). ^#^ Pairwise comparison revealed a significant difference between legs with asymptomatic varicose veins and legs with symptomatic varicose veins (*p* = 0.024). ^†^ Pairwise comparison revealed a significant difference between legs without varicose veins and legs with symptomatic varicose veins (*p* = 0.004). ^‡^ Pairwise comparisons revealed significant differences between legs with asymptomatic varicose veins and legs with symptomatic varicose veins (*p* = 0.02) and between legs without varicose veins and legs with symptomatic varicose veins (*p* = 0.026). Abbreviations: ASV, absolute stroke volume; GSV, great saphenous vein; MF, mean flux; MV, mean velocity; PV, popliteal vein; RF, regurgitation fraction.

**Table 4 diagnostics-12-00118-t004:** Comparison of QFlow findings for legs with varying levels of varicose vein symptom severity.

	No Symptoms *n* = 42	Discomfort or Edema *n* = 28	Skin Changes or Ulcers *n* = 18	*p*
Imaging findings				
Compression of iliocaval territory	11 (26.2)	7 (25.0)	7 (38.9)	0.54
Deep vein thrombosis	2 (4.8)	3 (10.7)	3 (16.7)	0.32
GSV dilatation *	16 (38.1)	22 (78.6)	13 (72.2)	<0.001
GSV to PV diameter ratio ^#^	0.52 (0.43, 0.75)	0.61 (0.47, 0.75)	0.68 (0.62, 0.91)	0.04
QFlow data findings				
External iliac vein				
RF (%)	0.00 (0.00, 0.00)	0.00 (0.00, 0.11)	0.00 (0.00, 0.09)	0.27
ASV (mL)	2.34 (1.85, 5.32)	3.71 (2.49, 5.13)	3.29 (2.51, 4.82)	0.69
MF (mL/s)	2.82 (2.01, 5.95)	3.37 (2.44, 6.69)	3.41 (2.51, 4.30)	0.62
MV (cm/s)	6.61 (4.56, 8.48)	7.51 (4.33, 9.38)	7.32 (4.79, 10.76)	0.72
Femoral vein				
RF (%)	0.00 (0.00, 0.00)	0.00 (0.00, 0.00)	0.00 (0.00, 0.00)	0.56
ASV (mL)	1.13 (0.82, 1.86)	1.11 (0.67, 1.75)	1.18 (0.90, 1.83)	0.79
MF (mL/s)	1.32 (0.94, 1.98)	1.34 (0.71, 2.06)	1.55 (0.89, 2.08)	0.93
MV (cm/s)	3.60 (2.61, 5.18)	3.90 (2.68, 6.27)	5.19 (3.25, 6.44)	0.13
Popliteal vein				
RF (%)	0.00 (0.00, 0.00)	0.00 (0.00, 0.00)	0.00 (0.00, 0.00)	0.78
ASV (mL)	0.80 (0.46, 1.10)	0.69 (0.39, 1.23)	0.85 (0.53, 1.50)	0.45
MF (mL/s)	0.85 (0.51, 1.14)	0.83 (0.43, 1.30)	1.09 (0.49, 1.77)	0.51
MV (cm/s)	1.70 (1.26, 2.31)	1.76 (1.19, 3.66)	2.41 (1.41, 3.19)	0.45
Great saphenous vein				
RF (%) ^†^	2.08 (0.00, 33.77)	0.00 (0.00, 4.31)	0.00 (0.00, 2.32)	0.03
ASV (mL) ^‡^	0.25 (0.16, 0.47)	0.33 (0.20, 0.63)	0.49 (0.27, 0.93)	0.03
MF (mL/s) ^§^	0.22 (0.08, 0.43)	0.37 (0.20, 0.82)	0.54 (0.23, 1.03)	0.02
MV (cm/s)	1.46 (0.55, 2.80)	2.43 (1.16, 3.43)	2.38 (0.94, 5.59)	0.12

* Pairwise comparison revealed that legs with no symptoms were different from legs with dis-comfort or edema and legs with skin changes or ulcers. ^#^ Pairwise comparisons revealed a significant difference between legs with no symptoms and legs with skin changes or ulcers (*p* = 0.04). ^†^ Pairwise comparison revealed no significant difference. ^‡^ Pairwise comparison revealed a significant difference between legs with no symptoms and legs with skin changes or ulcers (*p* = 0.03). ^§^ Pairwise comparison revealed a significant difference between legs with no symptoms and legs with skin changes or ulcers (*p* = 0.046). Abbreviations: ASV, absolute stroke volume; GSV, great saphenous vein; MF, mean flux; MRI, magnetic resonance imaging; MV, mean velocity; PV, popliteal vein; QFlow, quantitative flow; RF, regurgitation fraction.

## Data Availability

Restrictions apply to the availability of these data. Data was obtained from Chiayi Chang Gung Memorial Hospital and are available from the authors with the permission of Chiayi Chang Gung Memorial Hospital.

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
