# Peer review of "Blood Flow Analysis of the Great Saphenous Vein in the Su-Pine Position in Clinical Manifestations of Varicose Veins of Different Severities: Application of Phase-Contrast Magnetic Resonance Imaging Data"

_diagnostics, 2022, doi:10.3390/diagnostics12010118_

Round 1
Reviewer 1 Report
Summary:
In this manuscript, authors evaluated investigated changes in lower-limb venous blood flow using phase-contrast magnetic resonance angiography. They showed that legs without varicose veins exhibited a lower ASV and MF for the GSV compared with legs with symptomatic varicose veins. In addition, legs with asymptomatic varicose veins also exhibited a lower MF for the GSV compared with legs with symptomatic varicose veins, indicating increased blood flow rate and blood flow volume in the GSV were associated with symptomatic varicose veins.
Overall comments:
The work in this manuscript promotes the understanding of the link between blood flow data and the symptom severity of lower-limb varicose veins using phase-contrast magnetic resonance angiography. The data revealed the blood flow rate and blood flow volume in the GSV were associated with symptomatic varicose veins. However, from what the authors showed in Fig.3 and Table 4, the relationship between the blood flow rate/blood flow volume in the GSV and symptom severity is not so clear. Their data suggested no significant differences were observed between legs with discomfort or edema and legs with no symptoms, and between legs with discomfort or edema and legs with skin changes or ulcers. It seems that there may exist a threshold effect between blood flow data and the symptom severity of varicose veins, where only significant differences can be observed beyond a specific point of symptom severity. This evaluation may need a more detailed description on symptom severity, rather than only ranked as legs with discomfort or edema and legs with skin changes or ulcers. However, their conclusion is “the values of the GSV to PV diameter ratio, GSV ASV, and GSV MF may increase gradually with increased varicose vein progression (from legs with no symptoms to legs with discomfort or edema, and, finally, to legs with skin changes or ulcers)”, while this “gradually” increased effect can’t be supported by their data. Therefore, more related discussion and modifications on the text are needed.
Author Response
The work in this manuscript promotes the understanding of the link between blood flow data and the symptom severity of lower-limb varicose veins using phase-contrast magnetic resonance angiography. The data revealed the blood flow rate and blood flow volume in the GSV were associated with symptomatic varicose veins. However, from what the authors showed in Fig.3 and Table 4, the relationship between the blood flow rate/blood flow volume in the GSV and symptom severity is not so clear. Their data suggested no significant differences were observed between legs with discomfort or edema and legs with no symptoms, and between legs with discomfort or edema and legs with skin changes or ulcers. It seems that there may exist a threshold effect between blood flow data and the symptom severity of varicose veins, where only significant differences can be observed beyond a specific point of symptom severity. This evaluation may need a more detailed description on symptom severity, rather than only ranked as legs with discomfort or edema and legs with skin changes or ulcers. However, their conclusion is “the values of the GSV to PV diameter ratio, GSV ASV, and GSV MF may increase gradually with increased varicose vein progression (from legs with no symptoms to legs with discomfort or edema, and, finally, to legs with skin changes or ulcers)”, while this “gradually” increased effect can’t be supported by their data. Therefore, more related discussion and modifications on the text are needed.
Response:
(1) In this retrospective study, the grouping was mainly based on medical records, but most clinicians did not describe the clinical symptoms according to CEAP classification system. Another problem is that the number of cases evaluated with PC-MRA for lower extremity veins is still low. Therefore, in the classification of symptom severity, among the legs with varicose veins, we used the legs without symptoms, legs with discomfort or edema, and legs with skin changes or ulcers, which can include a larger range of subgroups, for our analysis. The purpose of this grouping method is to cover and classify the clinical symptoms recorded in the medical records as correctly as possible.
(2) “gradually” is just our speculation from the results (Table 4 and Figure 3), and there is really no strong evidence to support this speculation. However, among the legs with varicose veins, if the GSV ASV or GSV MF in the legs with discomfort or edema is significantly higher than that in the legs without symptoms, or if the GSV ASV or GSV MF in the legs with discomfort or edema is significantly lower than that in legs with skin changes or ulcers, or if both are present, then we tend to presume that there is a threshold effect between blood flow data and the symptom severity of varicose veins.
However, since the word “gradually” may cause controversy, we have deleted it from the revised manuscript.
We thank you for your comments.

Reviewer 2 Report
I read the paper sent to me for review with great interest. I have a few comments.
The works assessing the dynamics of changes in the venous system retrospectively have many limitations, including not taking into account behavioral factors, e.g. type and type of activity, family predispositions and many demographics. For example, the QoL of the questionnaire can also be used prospectively. An additional disadvantage is the lack of Doppler ultrasound evaluation in the control group, because the lack of varicose veins does not exclude GSV failure.
When it comes to changes in the volume of flow, I would recommend the authors refer to the pathophysiology of venous circulation and such laws as, for example, Bernoulli's law. The recumbent position is not appropriate for the haemodynamic assessment, e.g. of reflux. This could be done via the Valsalva Probe, for example. I don't understand what the authors meant by "The results of this study suggest that the morphology of the GSV may be… .."
Also, the lack of a relationship with the size of varicose veins and ailments is well known. Moreover, all patients with CVI show improvement after night rest, e.g. reduction of edema. An additional limitation is the result of a single NMR measurement.
Author Response
I read the paper sent to me for review with great interest. I have a few comments.
Point 1: The works assessing the dynamics of changes in the venous system retrospectively have many limitations, including not taking into account behavioral factors, e.g. type and type of activity, family predispositions and many demographics. For example, the QoL of the questionnaire can also be used prospectively. An additional disadvantage is the lack of Doppler ultrasound evaluation in the control group, because the lack of varicose veins does not exclude GSV failure.
Response 1:
Indeed, the lack of consideration of behavioral factors, family history, and doppler ultrasound evaluation in the assessment of venous hemodynamic changes is one of the limitations of this study. We have added this information to our Limitations.
Point 2. When it comes to changes in the volume of flow, I would recommend the authors refer to the pathophysiology of venous circulation and such laws as, for example, Bernoulli's law. The recumbent position is not appropriate for the haemodynamic assessment, e.g. of reflux. This could be done via the Valsalva Probe, for example. I don't understand what the authors meant by "The results of this study suggest that the morphology of the GSV may be… .."
Response 2:
- Because this study aims to investigate the relationship between the lower extremity venous blood flow data measured by QFlow scan and the clinical manifestations, the inaccuracy of the venous reflux data is expected. Therefore, we did not try to compensate for the shortcomings caused by the supine position in other ways.
- We have changed “morphology” to “the extent of dilatation” in the Discussion.
Point 3. Also, the lack of a relationship with the size of varicose veins and ailments is well known. Moreover, all patients with CVI show improvement after night rest, e.g. reduction of edema. An additional limitation is the result of a single NMR measurement.
Response 3:
Only a single PC-MRA measure is indeed another limitation. However, during outpatient visits, the patient usually tells the clinician how they look when their symptoms are most severe. Therefore, it is unlikely that the symptoms were recorded incorrectly because they became less severe at the time of the visit. Therefore, we believe that the impact of this limitation should not be significant.
We thank you for your comments.